# A Deep Learning Image System for Classifying High Oleic Sunflower Seed Varieties

**DOI:** 10.3390/s23052471

**Published:** 2023-02-23

**Authors:** Mikel Barrio-Conde, Marco Antonio Zanella, Javier Manuel Aguiar-Perez, Ruben Ruiz-Gonzalez, Jaime Gomez-Gil

**Affiliations:** 1Departamento de Teoría de la Señal y Comunicaciones e Ingeniería Telemática, Universidad de Valladolid, ETSI Telecomunicación, Paseo de Belén 15, 47011 Valladolid, Spain; 2Agricultural Engineering Department, Federal University of Lavras, P.O. Box 3037, Lavras 37200-000, Brazil; 3Department of Electromechanical Engineering, Escuela Politécnica Superior, University of Burgos, Avda. Cantabria s/n, 09006 Burgos, Spain

**Keywords:** classification system, convolutional neural network, high oleic sunflower seed

## Abstract

Sunflower seeds, one of the main oilseeds produced around the world, are widely used in the food industry. Mixtures of seed varieties can occur throughout the supply chain. Intermediaries and the food industry need to identify the varieties to produce high-quality products. Considering that high oleic oilseed varieties are similar, a computer-based system to classify varieties could be useful to the food industry. The objective of our study is to examine the capacity of deep learning (DL) algorithms to classify sunflower seeds. An image acquisition system, with controlled lighting and a Nikon camera in a fixed position, was constructed to take photos of 6000 seeds of six sunflower seed varieties. Images were used to create datasets for training, validation, and testing of the system. A CNN AlexNet model was implemented to perform variety classification, specifically classifying from two to six varieties. The classification model reached an accuracy value of 100% for two classes and 89.5% for the six classes. These values can be considered acceptable, because the varieties classified are very similar, and they can hardly be classified with the naked eye. This result proves that DL algorithms can be useful for classifying high oleic sunflower seeds.

## 1. Introduction

Agriculture and the food industry are facing issues to meet the demand of future generations for higher food production and healthier food [1]. The global population is estimated to rise to 9.7 billion people by 2064 [2]. Oilseeds are extensively cultivated throughout the world because oilseed-derived products are used as inputs in several stages of the food supply chain. The worldwide harvest of oilseeds in 2021/2022 amounted to 619.17 million metric tons [3]. The main producers of sunflowers (*Helianthus annuus* L.) are Brazil, the United States, and China. The main oilseeds that are cultivated around the world are sunflower seeds, soybean, and rapeseed [3].

Edible oil extract from oilseeds contains compounds with many health benefits and quality attributes that are important for the food industry [4]. There are two main types of food oils: conventional oil and high oleic oil. Conventional oil contains high contents of linoleic acid, while high oleic oil has a content of oleic acid that is 80% or higher and very low levels of linoleic acid. High oleic oil is the most stable type of oil for industrial use [5]. Besides, there is evidence that high oleic oil is a healthy choice for use in food production [6,7]. The food industry needs a substitute for trans fatty acids (TFA), which are often used in fried snacks and bakery products and have been associated with health problems [8]. Zambelli concluded that high oleic sunflower oil is the optimal substitute for TFA, due to its fatty acid profile and good agronomic performance [9].

Seeds, oil, and oilcake derived from sunflowers have numerous components suitable for a healthy diet [10]. After the oil extraction process, the largest product obtained is the oilseed press cake, which can be processed for human consumption [11], and is used in animal feed [12]. Additionally, sunflower-derived products have important uses, such as energy generation and industrial production of detergents, adsorbents, paints, lubricants, and adhesives, among others [13,14,15,16].

Sunflower production has been increasing over recent years, in response to the demands of the food industry and as a useful cash crop for farmers. Plant breeding methods have been applied to develop sunflower varieties, improving oil content and quality for industrial purposes [17]. The agronomic characteristics of sunflower varieties have also been improved, so that varieties can be adapted to adverse growing environments [18,19]. There are sunflower varieties that can differ in oil quality, yield, and by-products, as a result of genetic modifications [20]. Sunflower seed varieties may be mixed during harvesting, transportation, and storage processes. Seed classification systems are therefore useful to trade intermediaries and the food industry, to ensure that sunflower seeds are suitable for industrial requirements.

The manual classification of seed varieties can be biased and time-consuming, since physical attributes of seeds are normally hard to differentiate with the human eye. Researchers have been applying conventional image processing to overcome this classification problem successfully. In general, conventional approaches include: (i) a preprocessing stage, where processes such as cropping, scaling, color conversion, noise filtering, and normalization are applied [21]; (ii) a segmentation stage, where thresholding or watershed techniques are applied [21,22]; (iii) a feature extraction stage, where shape, morphological, texture, and color features are extracted [23,24]; and (iv) a classification stage, where classical machine learning techniques are applied, in the form of algorithms such as Gaussian Naive Bayes, random forest (RF), support vector machines (SVM), k-nearest neighbors (kNN), and artificial neural networks (ANN) [23,24,25,26,27]. Nevertheless, those traditional approaches, based on computer vision and feature extraction, could not guarantee a high enough accuracy of classification in most cases.

Deep learning (DL) is a modern approach considered as a subset of machine learning. It uses ANN to model high-level abstractions of data as the data volume increases. DL algorithms can be divided into four categories: restricted Boltzmann machines (RBM), autoencoders, sparse coding, and convolutional neural networks (CNN), the last-mentioned being the most suitable algorithms in the classification of objects in images [28]. Various CNN models have been tested to classify seeds [29,30,31,32], and researchers have suggested modifications to some existing models [33,34]. Zhao et al. developed a system to detect soybean seed surface defects with DL [35]; Nie et al. and Qiu et al. applied DL to classify seeds using hyperspectral imaging [36,37]; Loddo et al. proposed a novel CNN model for seed species classification [38]; and, Gao et al. developed a CNN model for the classification and recognition of wheat varieties, through the combination of wheat plant images in the tilling season, images of wheat plants in the flowering period, and images of the wheat seed [39].

DL has become popular in seed classification [22]. Its popularity is due to its easy application and high seed classification accuracy. It requires no further image processing despite preprocessing [40] and, in general, DL offers better results than classical machine learning approaches [41]. Lin et al. compared DL models with traditional hand-engineered approaches for rice kernel classification. The accuracy values of traditional approaches varied between 89.1% and 92.1%, while DL models achieved 95.5% [42]. Koklu et al. compared the performance of ANN models, deep ANN models, and DL models, for the classification of five varieties of rice, achieving classification accuracy values of 99.87%, 99.95%, and 100%, respectively [43]. Mukasa et al. tested multivariate machine learning and DL models in the discrimination of triploid watermelon seeds from diploid and tetraploid seeds. They reported accuracy values of 69.5% and 84.3% for multivariate machine learning, and 95.5% in DL models [44]. Finally, Przybyło and Jabłoński, using DL models for the classification of oak acorns, achieved slightly higher accuracy values than those achieved by human experts making manual predictions [40].

Regarding sunflower seeds, to the best of our knowledge, there are two studies that applied CNN to classify sunflower seeds. Kurtulmuş, in 2021, achieved 95% accuracy using a CNN model to classify sunflower seeds of some varieties cultivated in Turkey [45]. Li et al. developed a custom CNN model to sort sunflower seeds from inert material (leaves, stones, and defective seeds), achieving accuracies between 97.33% and 99.56% [46].

The aim of the present study is to classify up to six different varieties of sunflower seeds using a CNN model, and later assess the accuracies of the classification model. The main novelty of this article with respect to the existing literature is the comparison of accuracy achieved when increasing the number of varieties from two to six. Moreover, the varieties chosen in this study are different from the ones chosen in other published articles, and those seeds are visually less similar than the ones employed in this article.

## 2. Materials and Methods

A sunflower seed classifier system using a CNN model was implemented. The following paragraphs describe the seeds to be classified, the image acquisition system, the CNN model that was implemented to classify the seed varieties, and the hardware equipment and software employed to perform all the underlying processing. Figure 1 shows a block diagram of the processing methodology applied in the present study.

### 2.1. Sunflower Seeds

The following six sunflower (*Helianthus annulus* L.) seed varieties were classified: Talissman, from RAGT Seeds Group (Ickleton, UK); MAS830 and MAS89, from Maïsadour Cooperative (Haut-Mauco, France); Kaledonia, from Caussade Semences Group (Caussade, France); Gibraltar, from Syngenta (Basel, Switzerland); and Orientes, from KWS SAAT SE & Co. KGaA (Einbeck, Germany). All the sunflower seeds employed in this article were harvested in the year 2020, and they were distributed to the authors by Sovena Oilseeds España (Brenes, Spain). The seeds were manually cleaned of foreign material such as dirt, char, leaf debris, dust, and stones. Immature and defective seeds were discarded before image acquisition and only ripe and flawless seeds were selected for further processing. Sunflower seeds were stored in plastic buckets and kept at room temperature (20–25 °C). Sample images of these varieties are shown in Figure 2.

### 2.2. Image Acquisition Setup

The image acquisition system comprised: (i) a Nikon D90 camera equipped with a 60 mm AFD Micro-Nikkor lens f/2.8 to acquire the images; (ii) a steel bracket to mount the camera; (iii) an LED ring light, 14 inches in diameter, with 180 LEDs and 36 watts of power, to illuminate the seeds; (iv) a circular reflective metal ring placed around the seeds for consistent lighting; (v) a remote shutter to avoid trepidation; and (vi) a grid to place groups of 24 sunflower seeds in a fixed position. The distance between the camera lens and the seeds was 32 cm. In the camera configuration: (i) the shutter speed was fixed to 1/200, because lower speeds produced trepidation and results were no better at higher speeds; (ii) the working aperture was fixed at f:10 because wider apertures gave a worse depth of field and some details of the object were lost, and narrow apertures yielded a worse resolution and softer details; (iii) the ISO sensitivity was fixed at 200, because higher ISO produced more noise and, at a lower ISO, the results were no better and could provoke trepidation; and (iv) the white balance was fixed in flash mode to avoid variations of color temperature. Images were saved in JPG format at 4288 × 2848 resolution, employing the configuration in the camera that minimized the quality losses in the RAW to JPG conversion. The components of the image system acquisition in the position of seed sample preparation and also in the position of image capture are shown in Figure 3.

A total of 42 photos, with each photo containing a total of 24 seeds (Figure 3d), were taken for each of the six sunflower seed varieties under study.

### 2.3. Splitting of Seed Images

As explained in the previous section, the employed acquisition setup took a photograph of 24 seeds at the same time, placed in a grid of four rows and six columns (Figure 3d). Thus, it was necessary to split each seed into a different image. This process was made easier due to the use of a grid for the placement of the seeds, being achieved by using a constant mask for each cell, since the seeds were always placed at the same location within a bounding square. In this stage, the input images with 4288 × 2848 resolution containing 24 sunflower seeds were split to obtain images of individual seeds, with 712 × 712 pixels for each one. In addition, the size of each image was later reduced to a resolution of 227 × 227 pixels to match the input size of the CNN classifier. The scaling of the images was done using the OpenCV resize function in Python, with a default interpolation method.

After this stage, a dataset containing 6000 seed images, 1000 from each variety, was generated and used in the subsequent classification process.

### 2.4. CNN Classifier

The CNN architecture employed for classifying sunflower seeds in this work was AlexNet, originally proposed by Krizhevsky et al. [47]. This particular model was chosen, after a careful review of the literature, due to it outperforming other existing models when tackling similar problems to the one in this work. 

Figure 4 shows the architecture of AlexNet, with all involved layers and connections, while Table 1 shows the main parameters of each layer of this architecture.

The CNN model that was implemented, AlexNet, was fitted with the parameters listed in Table 1, and trained to maximize the classification accuracy using the Adam optimizer, minimizing losses with the cross-entropy function. Six sunflower varieties were used to implement and test the classifier model, varying the number of classes applied from two to six: (i) two classes (MAS830 and MAS89); (ii) three classes (Kaledonia, MAS830, and MAS89); (iii) four classes (Kaledonia, MAS830, MAS89, and Talissman); (iv) five classes (Kaledonia, MAS830, MAS89, Orientes, and Talissman); (v) six classes (Gibraltar, Kaledonia, MAS830, MAS89, Orientes, and Talissman). Table 2 shows the specific parameters of the AlexNet CNN model used for each combination of classes.

The distribution of the dataset was randomly divided into three parts: training, validation, and test blocks. From the 1000 images of each variety, 700 images were used for training (70%), 100 for the validation (10%), and 200 for the testing (20%). This division of the dataset into training–validation–test was chosen following the same approach employed in other similar articles, for a more direct comparison and also for the sake of obtaining a unique set of parameters after training instead of several sets of parameters, which happens when using cross-validation.

### 2.5. Evaluation of Results

Based on the results, the following evaluation metrics were employed: accuracy, recall, precision, loss, F1-score, ROC (receiver operating characteristic) curve, AUC (area under ROC curve), and the normalized confusion matrix.

### 2.6. Hardware Equipment and Software

The hardware equipment for the processing was a workstation with Intel(R) Xeon(R) CPU at 2.00 GHz, and an NVIDIA Tesla T4 graphic card with 16 GB of memory.

All the software involved in the processing in this article was programmed in Python, language version 3. For programming all the software related to the AlexNet CNN model, the TensorFlow library was used, in version 2.9.2.

## 3. Results

Different experiments were performed to analyse the AlexNet CNN model with the configuration given in Table 2. These experiments were based on varying the number of classes applied to the model from two to six: (i) two classes (MAS830 and MAS89); (ii) three classes (Kaledonia, MAS830, and MAS89); (iii) four classes (Kaledonia, MAS830, MAS89, and Talissman); (iv) five classes (Kaledonia, MAS830, MAS89, Orientes, and Talissman); (v) six classes (Gibraltar, Kaledonia, MAS830, MAS89, Orientes, and Talissman).

Figure 5 shows five graphs, one per experiment, with both the accuracy and the loss evolution during the training and validation phases, depending on the epoch. The horizontal axis represents the number of epochs. The vertical axis represents the value of the accuracies (red and yellow) and the losses (green and blue) during the training and validation phases. The graph shows that: (i) initially, as the number of epochs increased, the loss decreased, and therefore the accuracy increased; and (ii) when the number of epochs reached four, the curves began to stabilize.

The test dataset was used to verify the configuration of the model. The results of the model during this test phase are shown in Table 3 for the different experiments. These results include values of recall, precision, F1-score, AUC, accuracy, and loss.

Besides the results shown in Table 3, the normalized confusion matrix for the different experiments was computed, to obtain visual information on the test results, as shown in Figure 6. In this figure: (i) the horizontal axis is the predicted results; (ii) the vertical axis is the true values per class; (iii) the diagonal shows the classification accuracy per class of the model; and (iv) the other cells represent the classification errors per class.

Figure 7 shows ROC curves and the AUC for each variety in each experiment. The horizontal axis shows the false positive rate, and the vertical axis is the true positive rate.

According to the results shown in Table 3, Figure 6 and Figure 7, it can be seen that (i) the MAS89 variety was the best recognized seed; (ii) the Gibraltar variety was the worst recognized seed; and (iii) the accuracy decreases with increasing number of classes, from 100% for two classes to 89.5% for six classes.

## 4. Discussion

In our study, a seed classification accuracy of 89.5% was achieved in the classification with DL of six different varieties of high oleic sunflower seeds, Talissman, MAS830, MAS89, Kaledonia, Gibraltar, and Orientes, when the training, validation, and testing was performed with 1000 images of each variety, and the resolution of each image was 227 × 227 pixels. When less than six varieties were used for classification, accuracies reached values always higher than 94%, due to the Gibraltar variety being more difficult to distinguish from the rest of the seeds.

The accuracy of DL classification of varieties of seeds from plants such as sunflowers [45,46], chickpeas [34], maize [33], rice [43], grass [32], and mixed plants [48] has been evaluated in the literature.

There are, to the best of the authors’ knowledge, only two studies in the literature on sunflower seed classification with DL [45,46].

In the first one, Kurtulmuş identified four sunflower seed varieties (Armada, Reina, Sirena, and Palanci) as shown in Figure 8b [45]. In that study, an approximate total of 1200 seeds per variety (a total of 4800 seed samples) were imaged. Different CNN architectures were trained and tested using three models (AlexNet, GoogleNet, and ResNet) and two learning algorithms (AdaDelta and SGD). In their experiments, there was not a significant difference between training and testing accuracies. All the models could obtain classification accuracies over 90%. The GoogleNet model yielded the highest accuracy, with a value of 95%, in distinguishing the sunflower seeds. AlexNet with AdaDelta, and AlexNet with SGD, also provided relatively good classification performances (accuracy of 93%, AUC = 0.96), requiring notably short training times (a little less than one hour) than the others (more than two hours). Moreover, no particular solver (AdaDelta and SGD) was more prominent in performance.

Our study, and the study by Kurtulmuş, reached very close high performance rates. In our study, the accuracy score achieved in discriminating between four varieties of seeds was 97.2% (AUC = 0.99%) and in the Kurtulmuş study, when using AlexNet models, was 93% (AUC = 0.96). There are some differences in the methodologies that can explain the small differences. Our optimizer, or solver, was Adam, in contrast with the solvers used by Kurtulmuş, which were Ada Delta and SGD. Another difference is that the four seed varieties of both studies are different, making the comparison between studies more difficult. Nevertheless, in both studies, the performance metrics observed from the experiments were promising, with the fact that visually similar sunflower varieties could be identified by the methods proposed. Overall, both studies showed that the identification of sunflower seeds was feasible using deep learning techniques.

Figure 8a shows the sunflower seed varieties classified in our study and Figure 8b shows the sunflower seed varieties classified in the study of Kurtulmuş. As can be seen in both figures, the seeds of both studies look similar, and it is therefore coherent that both studies reached similar accuracies.

In the second study in the literature on sunflower seed classification, Li et al. conducted comparative experiments to compare the performance of the SeedSortNet network with six representative CNN models, that included AlexNet. They classified sunflower and maize seeds, though for the detection of abnormal seed images composed of leaves, stones, defective seeds, and normal sunflower seeds [46]. Experimental results show that AlexNet achieved accuracy rates of 99% and SeedSortNet of 99.56%, on sunflower seed dataset classification. Nevertheless, their study is not comparable with our study, because abnormal seed images are not sunflower seeds and therefore are easier to classify.

Some studies applying DL methods for classifying seeds reported accuracy values higher than the values achieved in our study. Kotlu et al. reported 100% accuracy in the classification of five rice varieties [43], Eryigit and Tugrul reported 99.4% accuracy for six grass varieties [32], and Xu et al. reported values between 96.4% and 99.7% for five maize varieties [33]. These higher accuracies could be due to a greater differentiation between the classes to be classified in their studies. That reasoning is coherent, because the images of the seeds in Figure 8c,d,f, visually differ in shape, size, color, and/or texture more than the images of the seeds shown in Figure 8a.

Other studies in the DL classification of seeds reported accuracies similar to the values found in our study, but lower than the aforementioned articles. Concretely, Taheri-Garavand et al. reported a classification accuracy of 94.2% for five chickpea varieties [34], and Loddo et al. reported accuracy values between 93.4% and 97.4% for 23 different varieties of seeds [48]. Those relatively lower accuracy values could be due to two reasons. The first one could be due to a smaller differentiation between the classes to be classified in the studies. That reasoning is coherent, because the images of the seeds shown in Figure 8f are more similar in shape, size, color, and texture than the images of the seeds shown in Figure 8a. The second one could be the higher number of seed varieties to be classified. That reasoning is also coherent, because twenty-three different seed classes are presented in Figure 8g, whereas only four are presented in Figure 8a.

One of the limitations of our study is the relatively small number of seed varieties employed, including only six sunflower varieties. Despite other studies also using only four varieties, like Kurtulmuş [45], there is room for improvement by incorporating additional varieties. In fact, the varieties used by Kurtulmuş were all different from the ones used by our study, and other existing varieties could be used too.

Regarding the main strength of our study, we considered seed varieties very similar in appearance, which made the differentiation of the classes more difficult. Moreover, the performance comparison of the behavior of the classifier when changing the number of classes showed the difficulties when adding new varieties, due to their similarity to one of the previously existing classes.

The classifier developed in the present study can be applied in industrial applications to rapidly analyze seed samples, since it uses a low-cost image acquisition system and is a non-destructive method. As sunflower varieties can differ in oil quality, yield, and by-products, the classifier can ensure that sunflower seeds meet the industrial requirements.

## 5. Conclusions

In summary, DL algorithms can be used to classify high oleic sunflower seeds with high accuracy. The accuracy values of classification decreased when adding more classes, varying from 100% to 89.5%. The largest decrease was observed with six classes, where the similarities within varieties increased, specially between the varieties Gibraltar and Kaledonia. Future studies could increase classification accuracies with different approaches, to advance this research. Varying specific parameters of the CNN model, such as the number of varieties for classification and the size of the dataset, could be tested. Future studies could also employ multispectral or hyperspectral images of high oleic sunflower seeds.

## Figures and Tables

**Figure 1 sensors-23-02471-f001:**
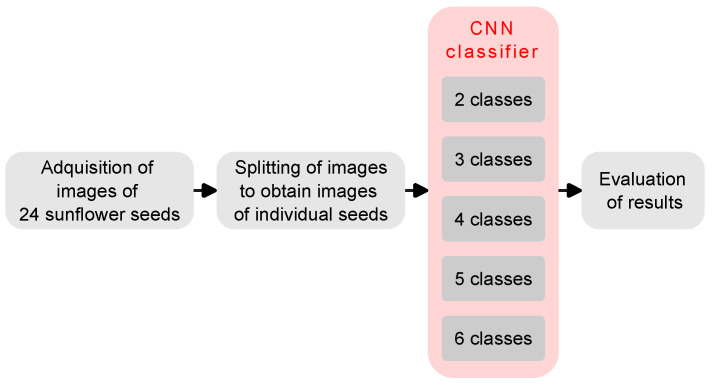
Block diagram of the processing methodology.

**Figure 2 sensors-23-02471-f002:**
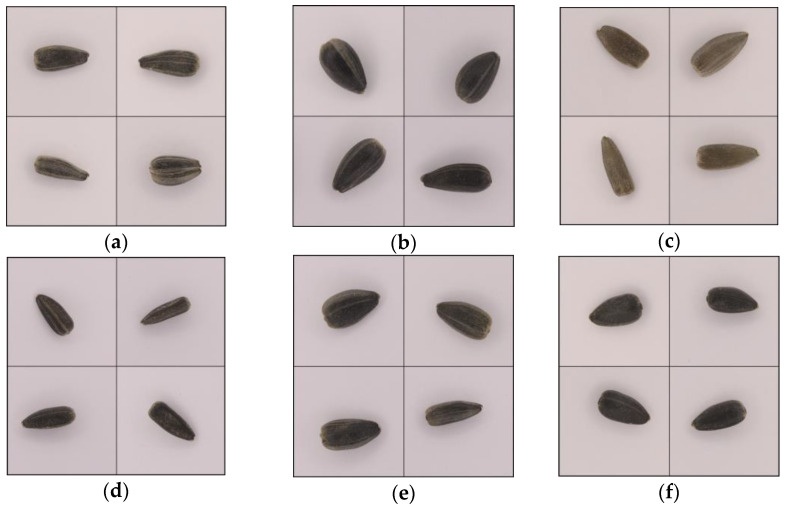
Example images of the six sunflower varieties used in the study: (**a**) Talissman, (**b**) MAS830, (**c**) MAS89, (**d**) Kaledonia, (**e**) Gibraltar, and (**f**) Orientes.

**Figure 3 sensors-23-02471-f003:**
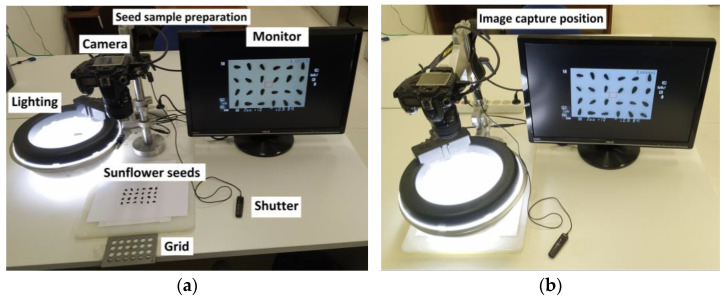
(**a**) Image system acquisition in the position of seed sample preparation, showing the camera, monitor, and lighting together with the grid used for positioning the seeds, and the remote shutter employed to trigger a photo; (**b**) image system acquisition in the position of image capture; (**c**) grid employed to place the 24 seeds into their position for acquisition; and (**d**) final position of the seeds after using the grid for their placement.

**Figure 4 sensors-23-02471-f004:**
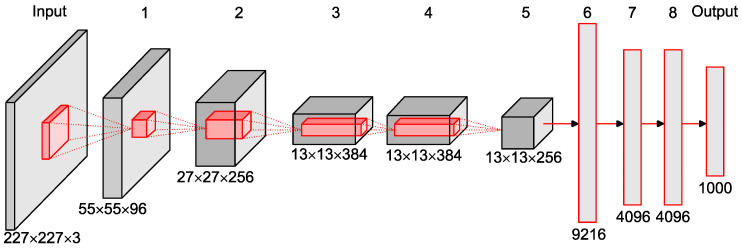
Architecture of the AlexNet CNN model.

**Figure 5 sensors-23-02471-f005:**
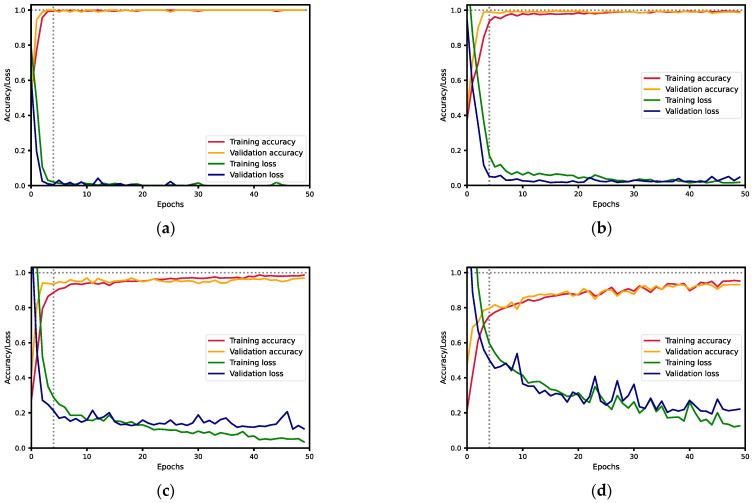
Curves of training and validation loss and accuracy for the different experiments: (**a**) two classes (MAS830 and MAS89); (**b**) three classes (Kaledonia, MAS830, and MAS89); (**c**) four classes (Kaledonia, MAS830, MAS89, and Talissman); (**d**) five classes (Kaledonia, MAS830, MAS89, Orientes, and Talissman); (**e**) six classes (Gibraltar, Kaledonia, MAS830, MAS89, Orientes, and Talissman).

**Figure 6 sensors-23-02471-f006:**
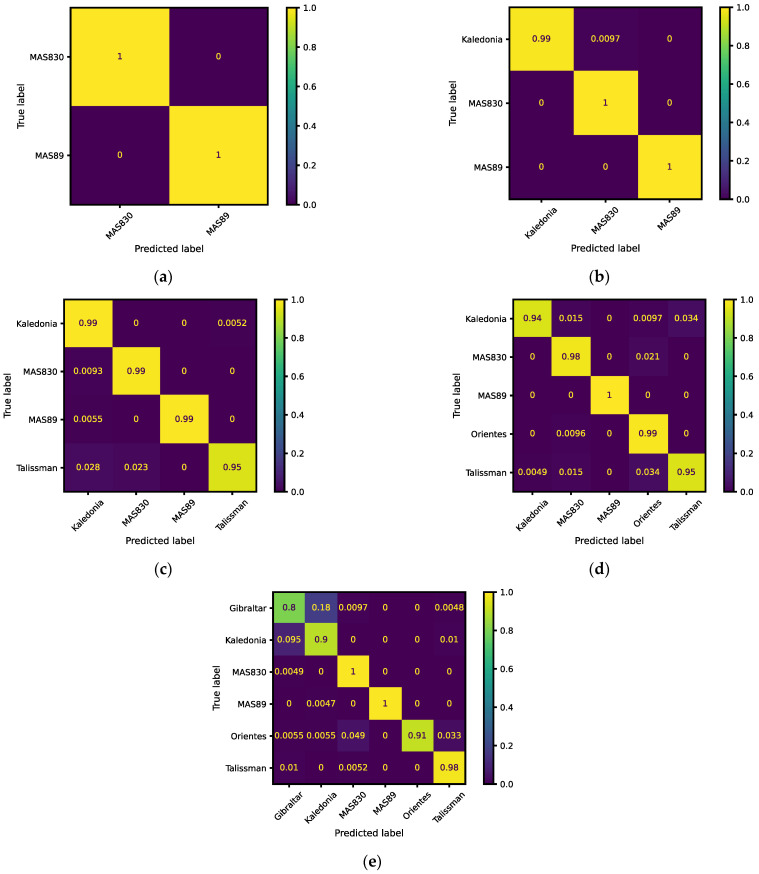
Normalized confusion matrix for the different experiments: (**a**) two classes (MAS830 and MAS89); (**b**) three classes (Kaledonia, MAS830, and MAS89); (**c**) four classes (Kaledonia, MAS830, MAS89, and Talissman); (**d**) five classes (Kaledonia, MAS830, MAS89, Orientes, and Talissman); (**e**) six classes (Gibraltar, Kaledonia, MAS830, MAS89, Orientes, and Talissman).

**Figure 7 sensors-23-02471-f007:**
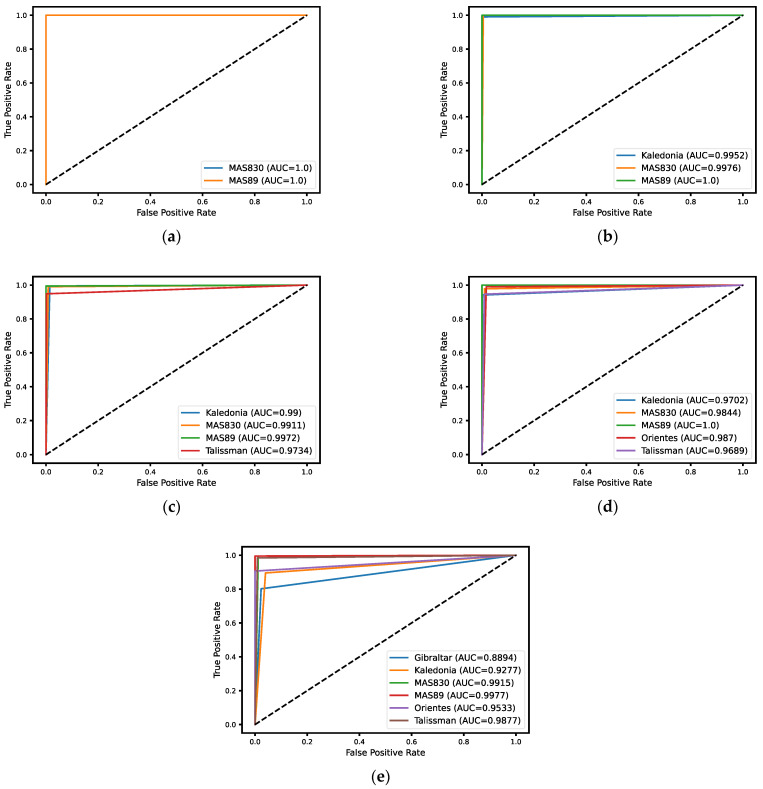
The ROC curves and AUC scores of the models with the highest accuracies for the different experiments: (**a**) two classes (MAS830 and MAS89); (**b**) three classes (Kaledonia, MAS830, and MAS89); (**c**) four classes (Kaledonia, MAS830, MAS89, and Talissman); (**d**) five classes (Kaledonia, MAS830, MAS89, Orientes, and Talissman); (**e**) six classes (Gibraltar, Kaledonia, MAS830, MAS89, Orientes, and Talissman).

**Figure 8 sensors-23-02471-f008:**
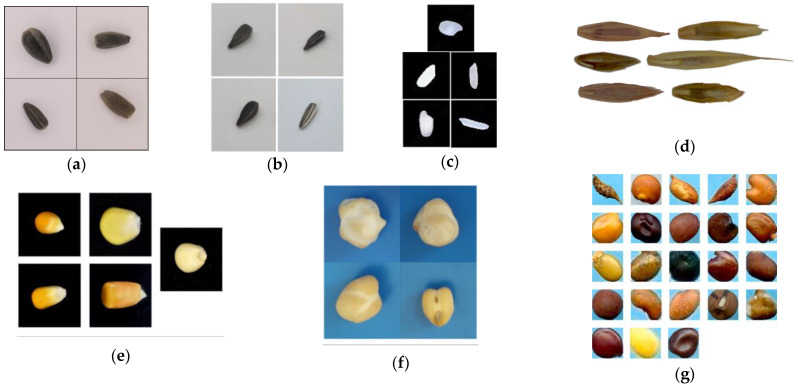
Images of varieties of seed classified in the literature. (**a**) Four high oleic sunflower seed varieties classified in our study. (**b**) Four sunflower seed varieties classified by Kurtulmus [45]. (**c**) Five rice varieties classified by Koklu et al. [43]. (**d**) Six grass varieties classified by Eryigit and Tugrul [32]. (**e**) Five maize varieties classified by Xu et al. [33]. (**f**) Four chickpea varieties classified by Taheri-Garavand [34]. (**g**) Twenty-three seeds classified by Loddo et al. [48]. All figures have been reproduced from their original papers with previous consent from the copyright holder.

**Table 1 sensors-23-02471-t001:** Specific parameters of each layer of the AlexNet CNN architecture.

Layer		Feature Map	Size	Kernel Size	Stride	Activation
Input	Image	1	227 × 227 × 3	-	-	-
1	Convolution	96	55 × 55 × 96	11 × 11	4	relu
	Max pooling	96	27 × 27 × 96	3 × 3	2	relu
2	Convolution	256	27 × 27 × 256	5 × 5	1	relu
	Max pooling	256	13 × 13 × 256	3 × 3	2	relu
3	Convolution	384	13 × 13 × 384	3 × 3	1	relu
4	Convolution	384	13 × 13 × 384	3 × 3	1	relu
5	Convolution	256	13 × 13 × 256	3 × 3	1	relu
	Max pooling	256	6 × 6 × 256	3 × 3	2	relu
6	FC	-	9216	-	-	relu
7	FC	-	4096	-	-	
8	FC	-	4096	-	-	relu
Output	FC	-	1000	-	-	softmax

**Table 2 sensors-23-02471-t002:** Specific parameters of the AlexNet CNN model.

Factor	Value
Image input size	227 × 227 × 3
Epochs	50
Optimizer	Adam
Loss function	Cross-entropy
Batch size	32
Validation size	30
Learning rate	0.00001
Dropout	0.5

**Table 3 sensors-23-02471-t003:** Classification results during the test phase.

Classes	Varieties	Recall	Precision	F1-Score	AUC	Accuracy	Loss
2	MAS830	1	1	1	1	1	0.00005
	MAS89	1	1	1	1		
3	Kaledonia	0.9903	1	0.9951	0.9952	0.992	0.02415
	MAS830	1	0.9896	0.9948	0.9976		
	MAS89	1	1	1	1		
4	Kaledonia	0.9948	0.9548	0.9744	0.9900	0.972	0.07590
	MAS830	0.9907	0.9770	0.9838	0.9911		
	MAS89	0.9945	1	0.9972	0.9972		
	Talissman	0.9486	0.9951	0.9713	0.9734		
5	Kaledonia	0.9417	0.9949	0.9676	0.9702	0.940	0.18497
	MAS830	0.9786	0.9581	0.9683	0.9844		
	MAS89	1	1	1	1		
	Orientes	0.9904	0.9406	0.9649	0.9870		
	Talissman	0.9466	0.9653	0.9559	0.9689		
6	Gibraltar	0.8019	0.8783	0.8384	0.8894	0.895	0.29350
	Kaledonia	0.8955	0.8182	0.8551	0.9277		
	MAS830	0.9951	0.9444	0.9691	0.9915		
	MAS89	0.9953	1	0.9977	0.9977		
	Orientes	0.9066	1	0.9510	0.9533		
	Talissman	0.9843	0.9543	0.9691	0.9877		

## Data Availability

Not applicable.

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
