# Peer review of "A Deep Learning Image System for Classifying High Oleic Sunflower Seed Varieties"

_sensors, 2023, doi:10.3390/s23052471_

Round 1

Reviewer 1 Report

The authors presented a method to classify sunflower seeds using DL algorithms. The paper has merit, but some points should be addressed.

In the introduction, could the authors contextualize the advantages of DL over the conventional imaging approaches? What are the limitations of the conventional imaging approach and of the classical machine learning approaches that justifies the use of DL?

Is the procedure that consist of dividing the data into training, validation and testing the best one?

Figure 4 and the discussion of Figure 4 should be improved. What does “ROC” stand for? What is the relevance of the area under the curve for evaluation of the method?

The authors mentioned that many seeds have been evaluated in the literature with DL and cited them. I believe that the paper would improve if the authors made a more detailed comparison between the presented results and the one from the literature.

Author Response

Please see the attachment file "Reviewer 1.pdf" the answers of your comments. 

Reviewer 2 Report

1)    The novelty of this paper is poor. Please add the novelty and detail contribution of the work under a subheading of introduction?

2)    Although references are used mostly recent however, please review more relevant works and find the research gap from there.

3)    A general high level block diagram or framework of complete work in this paper is required to add at the beginning of chapter 2 so that readers can follow up the entire technical work.

4)    image system acquisition provided in Figure 2 is required to improve further with more details about the image capture and more details of the hardware set up.

5)    Network architecture of AI techniques are missing. Please add them.

6)    What is the significance of CNN in your proposed work?

7)    You have claimed that your proposed model achieved 97.4% which higher that state of the art. However, how you validate your work if they don’t use the same data set that you have used in your case?

8)    Result section is too small. Further experimentation with results is expected.

9)    Under discussion you should add key strength, limitation and impact/significance of this work in real life scenarios.

10) Specific Future research directions are missing. Please add those at the end of conclusions.

Author Response

Please see the attachment file "Reviewer 2.pdf" the answers of your comments. 

Reviewer 3 Report

This study proposes to examine the capacity of Deep Learning algorithms to classify sunflower seeds. Its main contribution consists in automate the process of identifying the varieties.

The document is easy to read and follow.

The English needs minor spell checking.

The document is well supported with references.

The subject of the paper has great potential of application.

The proposed work main weakness is the lack of comparison of the proposed method with other state of the art methods. Also, it is not clear why authors chose the AlexNet CNN model and not another.

The document should be better reasoned related to the motivation and why the proposed system is needed. Finally, the results section should be deeply improved missing in the document some important performance metrics referred below.

There is no reference in the text to Figure 2. Please correct.

Authors should clarify in the text the reason to obtain images of individual seeds of 712 × 712 pixels for each one and the image input size in the CNN was 227x227x3. Also, authors should explain in more detail the image transformation from 712x712 to 227x227x3.

In the results section authors should include not only the accuracy and confusion matrix but also the f1-score, precision, recall, etc. This section should also include a comparison of the proposed solution with other state-of-the-art algorithms.

The discussion from line 199 to line 217 should be revised avoiding repetition of the same text and phrase structures.

The document is missing the conclusions section.

Author Response

Please see the attachment file "Reviewer 3.pdf" the answers of your comments. 

Round 2

Reviewer 2 Report

I would like to thank the authors for addressing my comments. I have no other comments.

Reviewer 3 Report

Since the main issues pointed out in the previous review were addressed by the authors I advise the manuscript to be accepted for publication.